TOPICAL REVIEW

# Cellular and molecular cross-talk in atrial fibrillation: The role of non-cardiomyocytes in creating an arrhythmogenic substrate

Zhenyu Dong[1,2] 🆔, Ruben Coronel[1,2] 🆔 and Joris R. de Groot[1,2]

[1]*Department of Clinical and Experimental Cardiology and Cardiothoracic Surgery, Heart Center, Amsterdam UMC, University of Amsterdam, Amsterdam, The Netherlands*
[2]*Amsterdam Cardiovascular Sciences, Heart Failure and Arrhythmias, Amsterdam, The Netherlands*

Handling Editors: Bjorn Knollmann & T Alexander Quinn

The peer review history is available in the Supporting Information section of this article (https://doi.org/10.1113/JP286978#support-information-section).

**Abstract figure legend** Illustration of cellular and molecular cross-talk in atrial fibrillation. Left: a schematic of cardiac tissue showing cardiomyocytes, fibroblasts, adipocytes, inflammatory cells and the coagulation system. Right: direct and indirect cross-talk between different cell types, with the impact of direct cross-talk on action potential (red continuous line represents cell coupling). Created using BioRender.com.

**Zhenyu Dong** completed medical training as a cardiac electrophysiologist in China (MD) and is pursuing a PhD under Professor de Groot and Professor Coronel, focusing on atrial fibrillation substrates and atrial cardiomyopathy. **Ruben Coronel** studied medicine at the University of Amsterdam (MD, 1981; PhD, 1988). He is an Associate Professor at the Heart Centre and Principal Investigator of the Experimental Cardiology Group at the Academic Medical Centre, Amsterdam, focusing on mechanisms and modulation of cardiac arrhythmias. From 2014 to 2021, he was a 'professeur invité' at the University of Bordeaux. He has published over 300 research papers, with an updated list available at www.ruben-coronel.nl.

R. Coronel and J. R. de Groot contributed equally to this work as secondary authors.

**Abstract** Atrial fibrillation (AF) is a complex arrhythmia. Various modulating factors influence its triggers and substrate. Fibroblasts, adipocytes, inflammatory cells and the coagulation system can disrupt cardiomyocyte function. Cardiomyocytes and fibroblasts release inflammatory cytokines that promote local and systemic inflammation, enhancing fibroblast activation and extracellular matrix deposition, leading to myocardial fibrosis. Fibrosis is essential for the induction of reentrant arrhythmias, including AF. Adipocytes contribute to arrhythmogenesis by secreting pro-inflammatory and pro-fibrotic factors, exacerbating inflammation and metabolic dysregulation. Inflammatory mediators activate the coagulation system, which augments this vicious cycle by producing factors promoting inflammation, fibrosis and arrhythmias at the same time as increasing the risk of thrombosis. Understanding these interconnected roles in the development and progress of the atrial arrhythmogenic substrate may point to potential novel therapeutic targets to stabilise or antagonise the atrial substrate and eventually prevent AF. This review examines the role of the interplay between cardiomyocytes, fibroblasts, adipocytes, inflammation and the coagulation system in contributing to the arrhythmogenic substrate for AF initiation and perpetuation.

(Received 30 September 2024; accepted after revision 14 February 2025; first published online 8 March 2025)

**Corresponding author** R. Coronel: Department of Experimental Cardiology, Heart Center, Amsterdam UMC, University of Amsterdam, Meibergdreef 9, 1105 AZ, Amsterdam, The Netherlands. Email: rubencoronel@gmail.com

## Introduction

Atrial fibrillation (AF) is the most common arrhythmia and is currently estimated to affect over 10.55 million patients in the USA and 11.7 million in the European Union (Krijthe et al., 2013; Noubiap et al., 2024). The pathophysiological mechanism of AF is formed by the arrhythmogenic electrical and structural changes within the atrium, pulmonary vein myocardium, and autonomic nervous system (McCauley et al., 2024). AF is initiated and maintained through a combination of triggers, such as early afterdepolarisation (EAD), delayed afterdepolarisation (DAD) and automaticity (Coumel, 1994; Nattel et al., 2020). These factors interact with the atrial substrate, characterised by changes in the effective refractory period, conduction slowing and fibrosis, to create a reentrant circuit that perpetuates the arrhythmia (Nattel et al., 2020). The arrhythmogenic substrate of AF includes pathological changes in atrial cardiomyocytes and is modulated by various cell types and regulatory factors beyond cardiomyocytes (McCauley et al., 2024). The combination of these changes is referred to as the atrial interactome (Nattel et al., 2020). In this review, we discuss emerging insights into the components of the interactome related to the genesis of AF.

Cardiomyocytes constitute three-quarters of myocardial tissue volume but less than half the total cell count (Vliegen et al., 1991). Other non-cardiomyocytes affecting the structural and functional properties of the heart that may lead to AF include (myo)fibroblasts, adipocytes, inflammatory cells, vascular smooth muscle cells, endothelial cells and neurons (Grandi et al., 2023). Fibrosis, characterised by an accumulation of extracellular matrix (ECM), arises from the multiplication of fibroblasts and their transformation into myofibroblasts that express $\alpha$-smooth muscle actin (Nattel, 2017). Increased production of ECM may hamper normal cardiac electrical and mechanical function, cause conduction heterogeneities, and stabilise and anchor reentrant circuits, thereby perpetuating AF (Nattel, 2017). Adipocytes and inflammatory cells exacerbate this process at the same time as directly affecting cardiomyocytes (Dobrev et al., 2023; Ernault et al., 2021; Mahajan et al., 2018). Endothelial–mesenchymal transition has been identified as a mechanism underlying profibrotic remodelling in AF (Saljic et al., 2022; van den Berg et al., 2021). Cardiac neurons, the autonomic nervous system and the coagulation system are also closely associated with the pathophysiology of AF (Chen et al., 2014; Spronk et al., 2017). Here, we discuss the role of non-cardiomyocyte-derived mediators involved in structural and electrical arrhythmogenic remodelling in AF.

Cross-talk describes various forms of intercellular communication. Anatomical cross-talk between two cell types refers to the abnormal presence of one cell type in the vicinity of another, resulting in alterations to the function of the affected cell type (Pyman et al., 2025). An example of anatomic cross-talk is myocardial fibrosis, caused by excessive ECM deposition through fibroblasts and other mesenchymal cells that may alter the myocardial activation pattern from homogeneously anisotropic (e.g. a homogeneously higher conduction velocity parallel than perpendicular to the fibre direction) to heterogeneously

anisotropic (Spach & Barr, 2000). Paracrine and autocrine signalling are forms of indirect cross-talk where signalling molecules released by a cell act on nearby cells or the signalling cell itself (Long, 1996; Sid-Otmane et al., 2020). By contrast, endocrine signalling involves circulating hormones or signalling molecules in blood that reach distant target cells (Sid-Otmane et al., 2020) (Fig. 1 and Table 1). Direct cross-talk is defined here as the physical coupling between two cell types that allows them to exchange matter and/or ions to propagate current or transmit mechanical forces (Rook et al., 1992). An example of direct cross-talk is the direct contact between cardiomyocytes and adipocytes, leading to electrophysiological changes in cardiomyocytes (Morrissette-McAlmon et al., 2024).

As a result of space constraints, we primarily discuss the interaction among cardiomyocytes, fibroblasts, adipocytes, inflammation and components of the coagulation system. Given the complexity of these interactions, we concentrate on the direct and indirect cross-talk involving cardiomyocytes in the context of AF and the factors contributing to it (Fig. 2). Table 2 summarises the structural and electrophysiological effects of representative individual factors involved in arrhythmogenic cross-talk.

## Indirect cross-talk

**Fibrosis and fibroblasts in AF.** Atrial fibrosis is a key pathophysiological factor in AF (McCauley et al., 2024). It plays a central role in causing conduction delays and activation blocks, which are essential for initiating the arrhythmia and stabilising the reentrant drivers that sustain it (Dzeshka et al., 2015; Haïssaguerre et al., 2016). Collagen deposition constitutes an anatomic cross-talk, where interactions between cardiac fibres and the ECM modulate myocardial conduction properties (Spach & Barr, 2000; Spach & Dolber, 1986). In the infarcted ventricular papillary muscle, 'zigzag' activation paths over branching cardiomyocyte bundles were observed, with fibrotic tissue separating these bundles and contributing to the inhomogeneous activation (de Bakker et al., 1990, 1993). Similar interactions are also observed in the atrium, where fibrosis significantly changes rotor dynamics (Roney et al., 2016). The presence of thick interstitial collagen within the diseased left atrium is associated with increased longitudinal conduction velocity but paradoxically longer activation times (Krul et al., 2015). Under pathological conditions, the normally low number of fibroblasts can increase through proliferation and the transformation of several other cell types, including

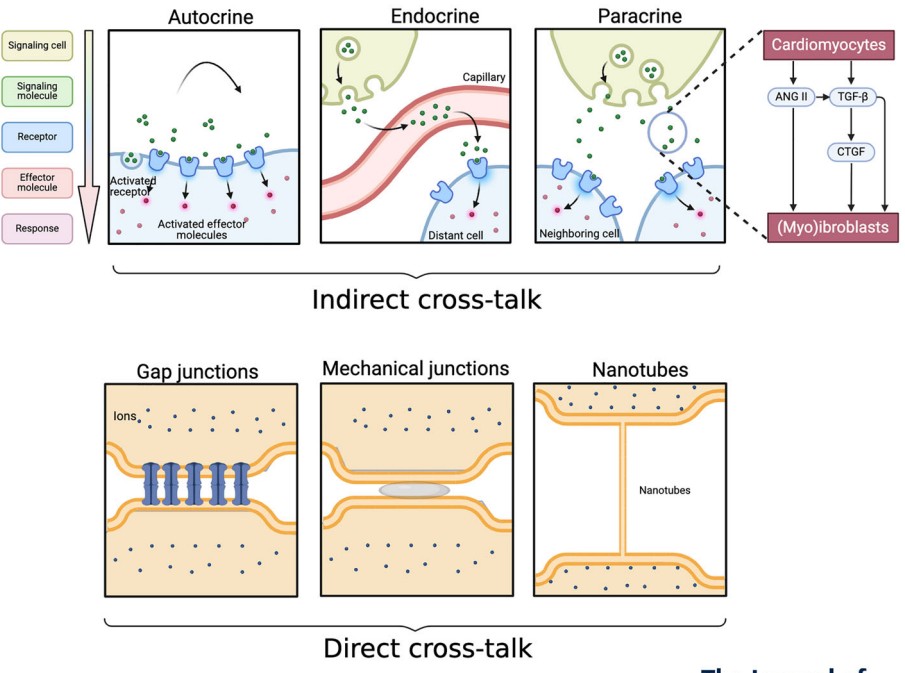

**Figure 1. Different modes of cell cross-talk**
Different modes of cell cross-talk include indirect methods, such as autocrine, endocrine and paracrine signalling, and direct methods, including nanotubes, mechanical interactionsd gap junctions. Right: interactions between signalling molecules, using transforming growth factor (TGF)-$\beta$, angiotensin II (Ang II) and connective tissue growth factor (CTGF) as examples (Chen et al., 2000; Tsai et al., 2011).

**Table 1. Types and examples of cellular and molecular cross-talk in atrial fibrillation.**

| | Type | Description | Example | References |
|---|---|---|---|---|
| Direct cross-talk | Gap-junction | Direct connections between adjacent cells, allowing exchange of ions and small molecules | Coupling between cardiomyocytes and myofibroblasts affects cardiomyocyte electrical activity | (Chilton et al., 2007) |
| | Mechanical junctions | Physical links between cells, providing structural support and signalling pathways | Myofibroblast tension on cardiomyocytes may impair conduction via mechanosensitive channel activation | (Thompson et al., 2011) |
| | Membrane nanotubes | Tubular structures that facilitate the transfer of molecules between cells | Membrane nanotubes between cardiomyocytes and fibroblasts | (He et al., 2011) |
| Indirect cross-talk | Paracrine | Cells release factors that affect nearby cells, influencing their function | Adipocytes secreting microRNA that impact cardiomyocytes | (Ernault et al., 2023) |
| | Autocrine | Cells respond to their own secreted factors, reinforcing their activity | Fibroblasts producing TGF-$\beta$ to enhance fibrosis | (Long, 1996) |
| | Endocrine | Signalling molecules released into the bloodstream affect distant cells, leading to systemic effects | Myeloperoxidase released by immune cells enter the bloodstream and subsequently affect cardiomyocytes | (Rudolph et al., 2010) |

monocytes and endothelial cells (Dzeshka et al., 2015; Krenning et al., 2010).

Various signalling molecules are involved in the interaction between fibroblasts and cardiomyocytes. Both fibroblasts and cardiomyocytes can secrete transforming growth factor (TGF)-$\beta$, and thus they may influence each other (Frangogiannis, 2022). Cardiomyocytes also secrete growth factors such as vascular endothelial growth factor, connective tissue growth factor (CTGF) and platelet-derived growth factor (Chen et al., 2017, 2000; Li et al., 2022; Tsoporis et al., 2012). These various growth factors play roles in the fibrotic signalling system and influence each other (Chen et al., 2000). Next, we discuss these factors based on their directionality of effect.

*Factors involved in indirect cross-talk from fibroblasts to cardiomyocytes*

*Growth factors.* A study by LaFramboise et al. (2007) showed that conditioned medium from cultured fibroblasts was added to neonatal cardiomyocytes, the cardiomyocytes exhibited hypertrophy and reduced contractile capacity. The medium contained various growth factors (LaFramboise et al., 2007). In mice with TGF-$\beta$ overexpression, AF inducibility was increased

and epicardial conduction velocity in atrial tissue was reduced, although action potential characteristics remained unchanged (Verheule et al., 2004). TGF-$\beta$ also caused a decrease in L-type Ca$^{2+}$ currents ($I_{CaL}$) and reduced Ca$_V$1.2 mRNA expression levels, where CaV1.2 mRNA encodes the $\alpha$1C subunit of the $I_{CaL}$ (Avila et al., 2007). Additionally, TGF-$\beta$ led to a reduction in the current densities of sodium currents, inward rectifier potassium currents ($I_{K1}$) and sustained outward potassium currents in neonatal rat atrial cardiomyocytes (Ramos-Mondragón et al., 2011).

*Factors involved in indirect cross-talk from cardiomyocytes to fibroblasts*

*Growth factors.* Injured cardiomyocytes secrete growth factors, including vascular endothelial growth factor and TGF-$\beta$, which in turn promote myofibroblast proliferation (Frangogiannis, 2022; Tsoporis et al., 2012). TGF-$\beta$ receptor stimulation in fibroblasts can activate various downstream pathways, which promote fibroblast proliferation, collagen synthesis and differentiation into myofibroblasts (Evans et al., 2003). TGF-$\beta$ induced CTGF in fibroblasts and promoted fibroblast proliferation, migration and ECM deposition (Chen et al., 2000)

**Table 2. Effects of representative non-cardiomyocytes on structural and electrophysiological changes in cardiomyocytes during atrial fibrillation.**

| Interaction factor | From | To | Structural changes | | Electrophysiology changes | | | | | | | | | | |
|---|---|---|---|---|---|---|---|---|---|---|---|---|---|---|---|
| | | | Fibrosis | Connexin | $I_{CaL}$ | $Ca^{2+}$ homeostasis | Other current | Conduction velocity | APD | ERP | RMP | EAD | DAD | AF inducibility | AF complexity |
| **Direct Cross-talk** | | | | | | | | | | | | | | | |
| Gap junctions | (Myo)fibroblast | Cardiomyocyte | | | | | | ↓ (18) | ↓ (20) | | ↑ (18) | | | | |
| Gap junctions | Macrophage | | | | | | | | ↓ (21) | | ↑ (25) | | | | |
| Gap junctions | Neutrophils | | | | | | | | ↑ (22) | | ↑ (22) | | | | |
| Gap junctions | Adipocytes | | | | | $Ca^{2+}$ Transients ↑ (12) | | ↓ (12) | ↑ (12) | | | | | | |
| **Indirect Cross-talk** | | | | | | | | | | | | | | | |
| TGF-β | (Myo)fibroblast / Platelets | Cardiomyocyte | Cardiomyocyte ↑ (1) | | ↓ (7) | | $I_{Na}$, $I_{K1}$, and $I_{Ksus}$ ↓ (16) | ↓ (19) | = (19) | | | | | ↑ (1) | |
| MicroRNA-1 | EAT | Cardiomyocyte | | | | | $I_{K1}$ ↓ (17) | | ↑ (17) | | ↑ (17) | | | | |
| FABP4 | EAT | | | | = (8) | Peak $Ca^{2+}$ ↓ (8) | | | = (8) | | | | | | |
| FXa | Coagulation | | ↑ (2) | | | | | | | | | | | ↑ (2) | ↑ (2) |
| Thrombin | Coagulation | | ↑ (2) | | | | | | | = (23) | | | | = (23) | |
| Activated human platelet products | Platelets | | | | ↑ (9) | | | | | | | ↑ (9) | ↑ (9) | | |
| TNF-α | Mainly from inflammatory cells | | ↑ (3) | Connexin 40↓ (5) | ↓ (10) | $Ca^{2+}$ Transients ↑ (13) $Ca^{2+}$ release ↑ (14) | | | ↑ (13) | | | | ↑ (10) | ↑ (3, 26) | |
| IL-1β | Mainly from inflammatory cells | | | Connexin 40/43↓ (6) | | $Ca^{2+}$ leak ↑ (15) | $I_{to}$ ↓ (15) | | ↑ (15) | | | | | | |
| IL-6 | Mainly from inflammatory cells | | | | ↑ or =* (11) | | | | | | | | | ↑ (27) | |
| IL-17A | Mainly from inflammatory cells | | | | | | | | | ↓ (24) | | | | ↑ (24) | |
| MPO | Mainly from neutrophils | | ↑ (4) | | | | | | | | ↑ (4) | | | | |

*Effects of various factors on atrial electrophysiology, depending on dose and exposure duration; $Ca^{2+}$ Transients: ↑ indicates prolonged duration; RMP: ↑ indicates depolarisation. Abbreviations: AF, atrial fibrillation; APD, action potential duration; DAD, delayed afterdepolarisation; EAD, early afterdepolarisations; EAT, epicardial adipose tissue; ERP, effective refractory period; $I_{CaL}$, L-type calcium current; $I_{K1}$, inward rectifier potassium current; $I_{Ksus}$, sustained outward potassium current; $I_{to}$, transient outward potassium current; $I_{Na}$, sodium current; RMP, resting membrane potential. References: 1 (Liu et al., 2020); 2 (Spronk et al., 2017); 3 (Saba et al., 2005); 4 (Al-Shama et al., 2023); 5 (Sawaya et al., 2007); 6 (Lazzerini et al., 2019); 7 (Avila et al., 2019); 8 (Lamounier-Zepter et al., 2009); 9 (de Jong et al., 2011); 10 (Lee et al., 2007); 11 (Ali et al., 2019); 12 (Morrissette-McAlmon et al., 2024); 13 (London et al., 2003); 14 (Zuo et al., 2018); 15 (Monnerat et al., 2016); 16 (Ramos-Mondragón et al., 2011); 17 (Yang et al., 2021); 18 (Yue et al., 2011); 19 (Verheule et al., 2004); 20 (MacCannell et al., 2007); 21 (Billur et al., 2023); 22 (Ward et al., 2006); 23 (Kondo et al., 2018); 24 (Fu et al., 2015); 25 (Hulsmans et al., 2017); 26 (Aschar-Sobbi et al., 2015); 27 (Li et al., 2023).

(Fig. 1). Fibroblasts exhibited a marked increase in ECM production in response to CTGF, a process that was entirely inhibited when a CTGF-neutralising antibody was introduced to the culture medium (Zhang et al., 2011). Platelet-derived growth factor-related signalling also promoted proliferation and ECM production in rat atrial fibroblasts (Jiang et al., 2016).

*Angiotensin II (ANG II).* ANG II released by cardiomyocytes also promoted fibrosis (Tsai et al., 2011). A medium conditioned from rapidly-paced atrial cardiomyocytes induced an activated myofibroblast phenotype that could be attenuated by ANG II receptor blockade (Burstein et al., 2007). The pacing of mouse atrial cardiomyocytes induced collagen expression in co-cultured mouse atrial fibroblasts through the secretion of ANG II and reactive oxygen species (Tsai et al., 2011). These effects could be mediated by the activation of CTGF through the TGF-$\beta$ signalling pathway (Tsai et al., 2011) (Fig. 1).

*Matrix metalloproteinases (MMPs) and tissue inhibitors of metalloproteinases (TIMPs).* MMPs and their tissue inhibitors TIMPs are also synthesised by cardiomyocytes and play a crucial role in regulating fibroblast function

(Kakkar & Lee, 2010). For example, TIMP-2 activated fibroblasts, transforming them into myofibroblasts (Ngu et al., 2014). However, at higher concentrations, TIMP-2 became inhibitory to this transformation process (Ngu et al., 2014).

*MicroRNA.* Several microRNAs produced by cardiomyocytes are associated with fibrosis (van den Berg et al., 2017, 2023). Thus, cardiomyocytes can influence fibroblast function by releasing microvesicles, including exosomes that carry microRNAs (Yang et al., 2018). In a rat model, microRNA-208a-containing exosomes promoted fibroblast proliferation and differentiation into myofibroblasts (Yang et al., 2018).

**Adipocytes in AF.** Adipocytes in the heart are primarily located within the epicardial adipose tissue (EAT), which is the visceral adipose tissue between the myocardium and the epicardium (Antonopoulos & Antoniades, 2017; Gawałko et al., 2023; Pyman et al., 2025). Adipocytes may be surrounded by dense fibrotic tissue (fibro-fatty infiltration) (Carpenter, 1962). Adipocyte infiltration into the myocardium can result in non-uniform anisotropic propagation of an activation wave front (anatomic

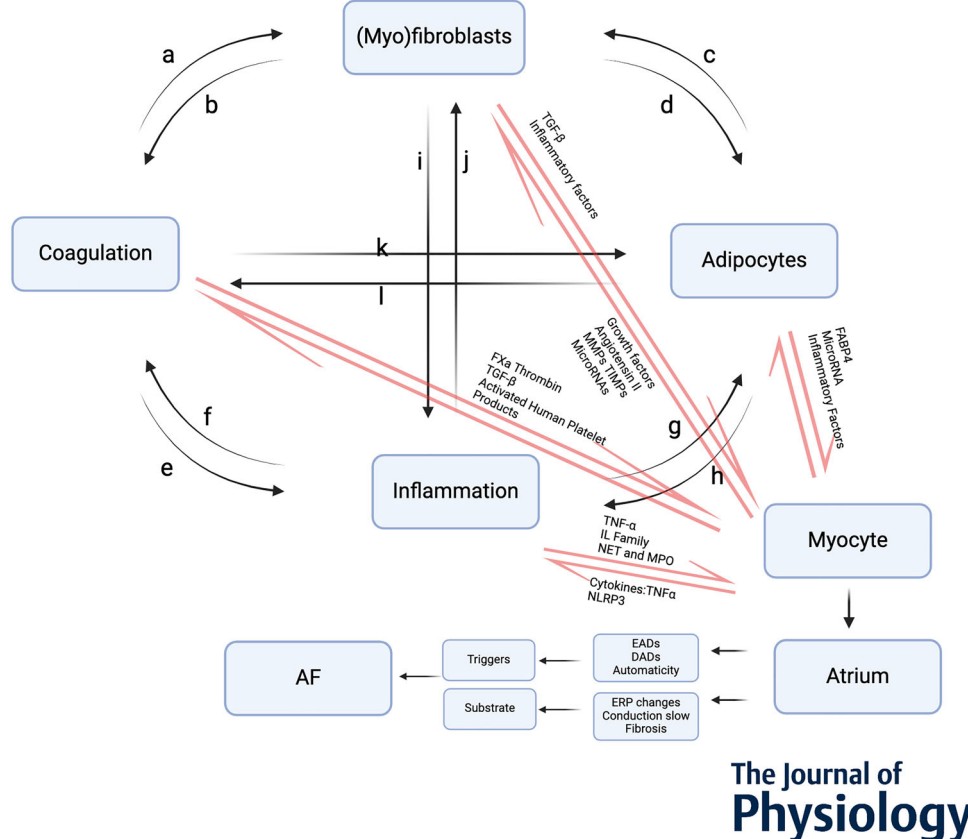

**Figure 2. The atrial interactome**
The interactome of non-cardiomyocytes and atrial cardiomyocytes. EAD, early afterdepolarisation; DAD, delayed afterdepolarisation; ERP, effective refractory period; AF, atrial fibrillation.

cross-talk) (de Bakker et al., 1993; Miles et al., 2023). The peri-atrial EAT is not separated from the underlying myocardium by a fascial layer and shares the same microvascular supply, thereby enabling reciprocal communication between them (Hatem & Sanders, 2014). EAT can secrete various soluble factors, including growth factors, adipokines, cytokines, bioactive lipids and extracellular vesicles (Ernault et al., 2021). EAT also secretes various interleukin (IL) family members and other inflammatory factors (Ernault et al., 2021). Ernault et al. (2022) demonstrated that culturing neonatal rat ventricular myocytes in the presence of EAT secretome reduced the expression of potassium inwardly rectifying channel subfamily J member 2 by 26%, leading to a 35% reduction in $I_{K1}$ and a subsequent decrease in the resting membrane potential to a less negative value. These electrophysiological changes promoted reentrant arrhythmias, as shown by computational modelling of the human left atrium partially covered with EAT (Ernault et al., 2022). This is consistent with findings in mouse models (Hatem & Sanders, 2014; Nalliah et al., 2020). The delayed rectifier potassium outward current was significantly reduced in rat cardiomyocytes upon exposure to the EAT secretome (Lee et al., 2013). Furthermore, incubating human induced pluripotent stem cell-derived cardiomyocytes with sheep EAT fragments for 24 h resulted in an extended duration of extracellular field potentials, indicating a lengthened action potential duration (APD) (Ernault et al., 2024; Nalliah et al., 2020). The signalling factors involved in the indirect cross-talk between adipocytes and cardiomyocytes are described below.

*Factors involved in indirect cross-talk from adipocytes to cardiomyocytes*

*Fatty acid-binding protein-4 (FABP4).* Adipokines released from EAT can modulate calcium dynamics in cardiomyocytes (Lamounier-Zepter et al., 2009). FABP4 is involved in transporting lipids to specific cellular compartments and reducing intracellular systolic peak $Ca^{2+}$ levels in rat cardiomyocytes (Lamounier-Zepter et al., 2009). Lamounier-Zepter et al. (2009) did not find evidence for an effect of FABP4 on APD and $I_{CaL}$.

*MicroRNA.* Extracellular vesicles secreted by EAT contain microRNAs that are upregulated compared to those in the secretome of subcutaneous adipose tissue (Ernault et al., 2023). These microRNAs promote arrhythmogenic conduction slowing (Ernault et al., 2023). In addition, microRNA-1 interacts with cardiac proteins, affecting ion channels such as the inwardly rectifying potassium channel 2.1 (Kir2.1), leading to less negative resting membrane potential and prolonged final repolarisation of the action potential in cardiomyocytes (Yang et al., 2021).

*Factors involved in indirect cross-talk from cardiomyocytes to adipocytes.* Additionally, the myocardium has been shown to modulate the EAT (Haemers et al., 2017). This was demonstrated indirectly in an obese sheep model, in which the volume of atrial adipose tissue significantly increased following AF induction (Haemers et al., 2017). Similarly, epicardial remodelling, driven by mechanical and haemodynamic stress, led to the expansion and fibro-fatty changes in peri-atrial EAT during chronic atrial myocardial remodelling (Suffee et al., 2020).

**Coagulation system in AF.** The coagulation system plays a critical role in cardiovascular diseases, including coronary heart disease and arrhythmias (Coronel et al., 1997; Watson et al., 2009). Following tissue injury, the extrinsic coagulation cascade is promptly triggered, resulting in the conversion of Factor X (FX) to activated FX, FXa. FXa then catalyses the transformation of prothrombin into thrombin, thereby commencing the formation of a blood clot (Mercer & Chambers, 2013). In addition to causing hemostasis, coagulation can convert extracellular procoagulant activity into intracellular signalling events via protease-activated receptors (Mercer & Chambers, 2013; Ramachandran et al., 2012).

AF has traditionally been considered to be a cause of hypercoagulability leading to ischaemic stroke rather than a consequence of left atrial enlargement and dysfunction creating a prothrombotic state characterised by blood stasis and endothelial impairment (Watson et al., 2009). However, this perspective has recently been challenged (Brambatti et al., 2014; de Jong et al., 2011; Fender et al., 2019). Stroke may be the first symptom of atrial cardiomyopathy and is mediated by hypercoagulability before the occurrence of blood stasis by AF (Brambatti et al., 2014; D'Alessandro et al., 2022). In concordance with this, the presence of a coronary blood clot caused ventricular arrhythmogenic effects (Coronel et al., 1997). Meta-analyses of longitudinal cohort studies have indicated that coagulation factors may be associated with the development of AF (Tilly et al., 2023). The impact of cardiomyocytes on the coagulation system is primarily considered to be associated with haemodynamic changes induced by the overall function of the myocardium (Watson et al., 2009). Accordingly, below, we primarily discuss the coagulation system's impact on cardiomyocytes.

*Factors involved in indirect cross-talk from coagulation to cardiomyocytes*

*FXa.* Multiple studies have shown that FXa is associated with atrial fibrosis and increased complexity of AF (Kondo et al., 2018; Matsuura et al., 2021; Spronk et al., 2017). Correspondingly, low molecular weight heparin, an inhibitor of FXa, reduced atrial fibrosis and decreased

the complexity of the AF substrate in goats (Spronk et al., 2017). Additionally, the FXa inhibitor rivaroxaban reduced the inducibility of AF by ∼50% in perfused mouse hearts (although not significant) without significantly altering the effective refractory period (Kondo et al., 2018).

*Thrombin.* FXa converts prothrombin into thrombin (Mercer & Chambers, 2013). Thrombin produced during coagulation promotes blood clotting and induces fibroblast activation and proliferation. Reduced thrombin generation decreases AF complexity and fibrosis in goats (Spronk et al., 2017). Specifically, maximum activation time differences and the fractionation index were significantly shorter in goats treated with low molecular weight heparin (Spronk et al., 2017). Research by Fender et al. (2020) suggested that thrombin promoted cardiac remodelling through protease-activated receptor 4.

*TGF-β.* Activated platelets released TGF-β, which enhanced ANG II-induced AF [see earlier section 'Angiontensin II (ANG II)']; in interaction with fibroblasts, activated platelets led to structural remodelling and fibrosis in the atria, thus facilitating AF (Liu et al., 2020; Verheule et al., 2004). Treatment with clopidogrel or the platelet-specific knockout of TGF-β mitigated Ang II-induced structural remodelling, atrial conduction disturbances, the inducibility of AF, and atrial inflammation and fibrosis in comparison to untreated mice (Liu et al., 2020).

*Activated human platelet products.* Activated human platelet products increased $I_{CaL}$ and intracellular $Ca^{2+}$, leading to APD prolongation and the occurrence of EADs and DADs in rabbit ventricular cardiomyocytes (de Jong et al., 2011). Aspirin prevented these potentially arrhythmogenic effects (Zakhrabova-Zwiauer et al., 2013).

**Inflammation in AF.** Increasing evidence indicates a close connection between systemic or local inflammation and AF (Dobrev et al., 2023). The proportions of immune cells within the atrium, including neutrophils and macrophages, significantly change during the progression of AF (Hulsmans et al., 2017; Yao et al., 2022). Inflammation may contribute to AF through electrical myocardial remodelling, but fibrosis is also a long-term consequence of inflammation (Dobrev et al., 2023; Hulsmans et al., 2023). Tumour necrosis factor (TNF)-α and the IL family can directly cause cardiomyocyte necrosis and apoptosis (Dobrev et al., 2023). This results in loss of cardiomyocyte function and decreased cardiac contractility, which increases AF risk (Dobrev et al., 2023). Inflammatory mediators also altered the electrophysiological properties of cardiomyocyte ion channels and $Ca^{2+}$ handling mechanisms (Dobrev et al., 2023; Fu et al., 2015; Lee et al., 2007) (Table 2). These changes may promote AF.

*Factors involved in indirect cross-talk from inflammatory cells to cardiomyocytes*

*TNF-α.* Transgenic mice overexpressing TNF-α in cardiomyocytes exhibited increased collagen accumulation, diminished contractile performance and weakened atrial contractility (Saba et al., 2005). Cardiomyocytes from these mice also exhibited a prolonged APD, with APD75 and APD90 increased by 14% and 28%, respectively (London et al., 2003). TNF-α is additionally associated with reduced atrial connexin 40 expression and heightened inducibility to pacing-induced AF (Saba et al., 2005; Sawaya et al., 2007) (Table 2). Knockout or pharmacological inhibition of TNF-α prevented exercise-induced atrial remodelling and reduced the inducibility of AF in mice (Aschar-Sobbi et al., 2015; Lakin et al., 2019).

TNF-α has also been linked to prolonged $Ca^{2+}$ transients in mice (London et al., 2003). *In vitro*, administration of TNF-α to rabbit pulmonary vein cardiomyocytes resulted in decreased $I_{CaL}$, increased amplitude of DAD, and elevated transient inward current and $Na^+/Ca^{2+}$ exchanger current (Lee et al., 2007). It also rapidly enhanced spontaneous $Ca^{2+}$ release in atrial cardiomyocytes through reactive oxygen species-mediated mechanisms (Zuo et al., 2018).

*IL family.* Components of the IL family affect the electrical substrate of AF (Monnerat et al., 2016). Elevated IL-6 signalling levels in patients were positively associated with AF, and selective blockade of IL-6 signalling reduced AF inducibility (Li et al., 2023). IL-1β prolonged the rat cardiac ventricular APD by decreasing transient outward potassium current (Monnerat et al., 2016). Consistently, IL-1β exposure induced a prolonged field potential duration in human-induced pluripotent stem cell-derived cardiomyocytes (Monnerat et al., 2016). The duration of the field potential can be used as a surrogate measure of the local activation recovery interval in the unipolar electrograms (Ernault et al., 2024). IL-1β also increased spontaneous diastolic sarcoplasmic reticulum $Ca^{2+}$ release in cardiomyocytes and decreased the expression of $Ca^{2+}$-handling proteins (Heijman et al., 2020; Monnerat et al., 2016; Szekely & Arbel, 2018). The effects of IL-1β on $Ca^{2+}$ handling proteins are probably amplified by IL-6, which induces reversible atrial electrical remodelling by downregulating the expression of connexin 40 and connexin 43 in mouse atrial cardiomyocytes (Lazzerini et al., 2019). It also directly changed $I_{CaL}$ (the effect varies depending on the dose and duration of exposure) and downregulated sarcoplasmic reticulum $Ca^{2+}$-ATPase activity and expression (Alí et al., 2019).

Increased IL-17A levels in the atrium were associated with the inducibility of AF in rats (Fu et al., 2015). Subsequently, rats treated with anti-IL-17A monoclonal antibodies showed significantly lower incidences and durations of AF (Fu et al., 2015). Additionally, anti-IL-17A treatment significantly increased the atrial refractory period and atrioventricular nodal refractory period (Fu et al., 2015).

*Neutrophil extracellular traps (NETs) and myeloperoxidase (MPO).* Experimental and clinical data has shown that NETs and MPO play a role in the pathogenesis of AF (He et al., 2023; Mołek et al., 2022; Rudolph et al., 2010). Interaction between NETs and cardiomyocytes was confirmed to contribute to AF progression (He et al., 2023). Proteomics revealed elevated MPO levels in the EAT secretome and left atrial tissue of persistent AF patients relative to those with paroxysmal or no AF (Meulendijks et al., 2023). Also, MPO could be detected in atrial EAT of patients prior to AF onset but not in patients who did not develop AF later (Meulendijks et al., 2023). Exposure of cultured neonatal ventricular rat cardiomyocytes to MPO made the cardiomyocyte resting membrane potential less negative, increased fibroblast numbers and elevated the expression of ECM genes (Al-Shama et al., 2023).

*Factors involved in indirect cross-talk from cardiomyocytes to inflammatory cells*

*Cytokines.* Under stress or following damage, cardiomyocytes release TNF-$\alpha$ and other cytokines, triggering inflammatory responses (Vinten-Johansen, 2004; Yu et al., 2012). This attracts immune cells to the damaged area, further exacerbating the inflammatory reaction (Vinten-Johansen, 2004). Subsequent AF, in turn, may generate an inflammatory response that initiates a vicious cycle (Dobrev et al., 2023). For example, stimulated cardiomyocytes produced 'cardiokines' that promoted macrophage production of MMP-9, IL-1$\beta$ and IL-6 (Li et al., 2012).

*NACHT, LRR and PYD domain containing protein 3 (NLRP3) inflammasome.* NLRP3 inflammasome was upregulated in atrial cardiomyocytes of patients with AF (Yao et al., 2018). Increased NLRP3 activity in atrial cardiomyocytes was linked with atrial fibrosis and higher AF inducibility in animal models (Heijman et al., 2020; Yao et al., 2018). Atrial macrophage infiltration may be driven by cardiomyocyte NLRP3 inflammasome activation, leading to the subsequent recruitment of macrophages to the atria (Dobrev et al., 2023). Additionally, several AF-promoting comorbidities that were demonstrated to be closely associated with AF progression also contributed to AF occurrence through NLRP3 inflammasome activation, including obesity

(Scott et al., 2021), chronic kidney disease (Song et al., 2023) and gut microbiota (Gawałko et al., 2022).

## Direct cross-talk

Direct cross-talk, an anatomical form of cross-talk, facilitates the exchange of currents, ions and other substances between cardiomyocytes and non-cardiomyocytes, enabling electrical coupling (Rook et al., 1992). The approximate resting membrane potential of atrial cardiomyocytes is–90 mV, whereas it is –30 mV in adipocytes (Bentley et al., 2014). Fibroblasts and macrophages have resting membrane potentials ranging from –10 to –35 mV (Hulsmans et al., 2017; Kamkin et al., 1999; Simon-Chica et al., 2023). Chilton et al. (2005) demonstrated that Kir2.1 in fibroblasts directly affects the resting membrane potential, bringing it closer to the potassium equilibrium potential. However, the resting membrane potential in fibroblasts exhibits notable heterogeneity (Simon-Chica et al., 2023). When non-cardiomyocytes electrically couple with cardiomyocytes, the membrane potentials of both cell types may shift, causing cardiomyocytes to depolarise slightly and non-cardiomyocytes to hyperpolarise (Ernault et al., 2021). Excessive depolarisation can partially inactivate fast sodium channels, slowing the action potential upstroke and reducing conduction velocity (Ernault et al., 2021). In severe cases, excessive depolarisation can fully inactivate these channels, making conduction reliant on the slower dynamics of $I_{CaL}$ (Ernault et al., 2021). However, a shift towards a more negative potential of the cardiomyocyte occurs when their intrinsic plateau potential is more positive than that of fibroblasts, leading to corresponding changes in repolarisation dynamics (Nattel, 2018). Direct cross-talk predominantly involves gap junctions, mechanical coupling and nanotubules (Gaudesius et al., 2003; Rook et al., 1992; Thompson et al., 2011).

**Gap junctions.** Cardiomyocytes are electrically coupled through gap junctions, enabling the myocardium to act as an electrical syncytium with subsequent synchrony in electrical activation and contraction (Rook et al., 1992). Gap junctions also exist between different cell types in the cardiac microenvironment (Hulsmans et al., 2017; Rook et al., 1992). Currently, reported interactions include those between cardiomyocytes and fibroblasts, between cardiomyocytes and macrophages, and between cardiomyocytes and adipocytes (Camelliti et al., 2004; Chilton et al., 2007; Hulsmans et al., 2017; Morrissette-McAlmon et al., 2024) (Fig. 2).

Electrotonic coupling between cardiomyocytes and fibroblasts through gap junctions can enhance phase 4 depolarisation, trigger automaticity, shorten APD, slow

conduction velocity and promote reentrant arrhythmias (MacCannell et al., 2007; Yue et al., 2011). However, data suggested that connexin 40 blockade alone was insufficient to antagonise these effects because, in cases of direct contact, only the simultaneous blockade of IL-6 and connexin 40 could reduce the functional changes induced by myofibroblasts (Johnson et al., 2023).

Inflammatory cells may also be directly coupled to cardiomyocytes through gap junctions (Billur et al., 2023; Hulsmans et al., 2017). Macrophages connected to spontaneously beating cardiomyocytes through connexin-43-containing gap junctions made the resting membrane potential of cardiomyocytes less negative, accelerated initial repolarisation and shortened the APD (Billur et al., 2023; Hulsmans et al., 2017). Upon adhesion to cardiomyocytes, migrating neutrophils also induced depolarisation of the cardiomyocyte resting membrane, leading to marked prolongation of action potentials (Ward et al., 2006). Neutrophils released cytotoxic oxidants and proteolytic enzymes that disrupted gap junctions and impaired myocardial cell function (Entman et al., 1992). These interactions disrupted cellular electrical stability in a heterogeneous manner, contributing to arrhythmogenic conditions.

Adipocytes may also form gap junctions with cardiomyocytes (Ernault et al., 2021). Studies have shown that human-induced pluripotent stem cell-derived cardiomyocytes, when directly co-cultured with human adipocytes, display notable electrophysiological alterations, including APD prolongation, reduced conduction velocity, increased conduction heterogeneity and prolonged $Ca^{2+}$ transients (Morrissette-McAlmon et al., 2024).

**Mechanical junctions.** Mechanical junctions and gap junctions together form the intercalated discs of cardiomyocytes (Sheikh et al., 2009). Similar to gap junctions, these mechanical junctions exist not only between cardiomyocytes, but also between fibroblasts and cardiomyocytes (Thompson et al., 2011). In co-cultures of fibroblasts and cardiomyocytes, the conduction velocity decreased (Thompson et al., 2011). However, the conduction velocity increased after applying contraction uncouplers and mechanical-sensitive ion channel blockers (Thompson et al., 2011).

**Membrane nanotubes.** Another potential way to influence direct cell-to-cell cross-talk is through membrane nanotubes (Rustom et al., 2004). Membrane nanotubes are thin connections enabling long-distance transport of $Ca^{2+}$, mitochondria and membrane-bound components, as observed between cardiomyocytes and fibroblasts *in vitro* and *in vivo* (Gaudesius et al., 2003; Gerdes & Carvalho, 2008; He et al., 2011) (Fig. 1).

## Integrated cross-talk

We have explored the interactions between non-cardiomyocytes and cardiomyocytes (red lines in Fig. 2). Figure 2 shows that the cross-talk can occur in a network of interactions. These individual interactions with cardiomyocytes lead to fibrosis, changes in electrical properties and $Ca^{2+}$ handling, and an increased pro-thrombotic state (Krishnan et al., 2021). The net effects of the combined (synergistic/antagonistic) interactions may contribute to an arrhythmogenic environment and may also involve cross-talk between non-cardiomyocytes (Krishnan et al., 2021) (black lines in Fig. 2). For example, activated fibroblasts secreted various inflammatory cytokines that promoted local inflammatory responses and triggered systemic inflammation through the bloodstream (McCauley et al., 2024) ('i' in Fig. 2). Subsequently, these activated inflammatory factors enhanced the coagulation process (Levi & van der Poll, 2010) ('f' in Fig. 2). In turn, thrombin also caused inflammatory responses (Levi & van der Poll, 2010; Strande & Phillips, 2009) ('e' in Fig. 2). Macrophages also expressed and released secreted phosphoprotein 1 (SPP1), which contributed to the promotion of fibrosis (Hulsmans et al., 2023) ('j' in Fig. 2).

Other examples of complex cross-talk centre around adipocytes. Adipocytes secrete various signalling factors, and these factors can diffuse into the adjacent atrial myocardium (Ernault et al., 2021). Extracellular vesicles derived from EAT and collected from patients with AF contained profibrotic cytokines and microRNAs, potentially delivering these factors to non-cardiomyocytes and increasing ECM synthesis (Shaihov-Teper et al., 2021) ('c' in Fig. 2). Adipocytes also released various inflammatory factors, triggering local inflammatory responses (Morrissette-McAlmon et al., 2024) ('h' in Fig. 2). These inflammatory factors could subsequently act on cardiomyocytes, altering their electrophysiological properties and increasing excitability and vulnerability, thus promoting AF (Dobrev et al., 2023). Conversely, anti-inflammatory and antifibrotic adipokines (omentin, apelin and adiponectin) are also secreted by adipocytes (Krishnan et al., 2021).

Cross-talk between adipocytes and the coagulation system adds to the complexity of interactions. Adipocytes can secrete plasminogen activator inhibitors, promote coagulation, and increase the risk of thrombosis (Kaji, 2016) ('l' in Fig. 2). Through this mechanism, EAT promoted the formation and stabilisation of atherosclerotic plaques (Kohler & Grant, 2000). Additionally, protease-activated receptor 4 promoted inflammatory responses through the NLRP3 inflammasome (Fender et al., 2020) ('e' in Fig. 2). Micro-thrombi could also provoke inflammation, contributing to the progression of fibrosis and further electrical

disturbances, creating positive feedback mechanisms maintaining AF (Kell et al., 2024) ('e' in Fig. 2). Thrombin produced during coagulation promoted blood clotting and acted on fibroblasts through its receptors, inducing their activation and proliferation (Altieri et al., 2018) ('a' in Fig. 2). Reduced thrombin generation was shown to mitigate AF-related fibrosis in AF goats (Spronk et al., 2017). Additionally, thrombin also acted directly on adipocytes, stimulating the secretion of inflammatory cytokines that may contribute to this fibrotic process (Strande & Phillips, 2009) ('k' and 'h' in Fig. 2).

In summary, the complex interactions of the above components may contribute to the substrate for AF. Although the balance between arrhythmogenic and anti-arrhythmic factors is delicate, the multidirectional and complex cross-talk offers the opportunity to control and mitigate arrhythmogenesis (Krishnan et al., 2021).

## Conclusions

In AF, the complex cross-talk among cardiomyocytes, non-cardiomyocytes and other regulatory factors contributes to an arrhythmogenic substrate (Goette et al., 2024). Current AF treatments, including anti-arrhythmic drugs, bradycardic agents, anticoagulants and ablation procedures, aim to restore or maintain sinus rhythm and prevent complications (McCauley et al., 2024). These treatments do not specifically address the AF substrate and the underlying cellular interactions. There remains a significant unmet need for therapeutic innovation targeted at disease-causing mechanisms. Understanding the intricate cellular and molecular cross-talk in AF may help identify novel personalised therapeutic targets. This approach, aiming at the multiple contributors to AF and its associated complications, has the potential to improve patient outcomes through the development of truly causal therapeutic strategies. Our review underscores how a complex network of cellular cross-talk involving inflammation, fibrosis, EAT and the coagulation cascade collectively contributes to an arrhythmogenic substrate for AF. A combination of EAT reduction via weight loss, cardiomyocyte-targeted anti-arrhythmic therapy, anti-inflammatory regimens, anti-fibrotic therapy, precision anticoagulation and selective electrical or mechanical uncoupling has the potential to more effectively combat AF and its consequences.

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

# Additional information

## Data availability statement

This study did not generate/analyse any datasets or code.

## Competing interests

J.dG has received research grant funding through his institution from Atricure Inc., Bayer, Boston Scientific, Daiichi Sankyo, Johnson & Johnson, Medtronic and Philips. He has also received honoraria and consultancy fees from Atricure Inc., Bayer, Berlin-Chemie, Daiichi Sankyo, Johnson & Johnson, Menarini, Medtronic, Novartis and Servier.

## Author contributions

Z.D drafted the manuscript. J.dG and R.C contributed by revising and reviewing the manuscript.

## Funding

This work is supported by a European Research Area for Health(ERA4Health) PERSUADE consortium grant, by the Information Technology for European Advancement (ITEA4) grant, number 21026, as well as by the CVON/Dutch Heart Foundation grant 01-002-2022-0118 for the Electrophysiological and Molecular Biomarkers in Atrial Fibrillation (EmbRACE) project.

## Acknowledgements

We acknowledge the use of OPENAI's ChatGPT for language refinement. However, we have thoroughly reviewed and verified all modifications.

## Keywords

atrial fibrillation, cross-talk, non-cardiomyocytes

## Supporting information

Additional supporting information can be found online in the Supporting Information section at the end of the HTML view of the article. Supporting information files available:

**Peer Review History**

