## [Peer Review History · The Journal of Physiology]

Cellular and Molecular Cross-talk in Atrial Fibrillation: The Role of Non-cardiomyocytes in Creating an Arrhythmogenic Substrate

Zhenyu Dong, Ruben Coronel, and Joris R. de Groot

DOI: 10.1113/JP286978

Corresponding author(s): Ruben Coronel (rubencoronel@gmail.com)

The following individual(s) involved in review of this submission have agreed to reveal their identity: Lisa Chilton (Referee #2)

Review Timeline:

Submission Date:	30-Sep-2024
Editorial Decision:	24-Oct-2024
Revision Received:	25-Jan-2025
Accepted:	14-Feb-2025

Senior Editor: Bjorn Knollmann

Reviewing Editor: T Alexander Quinn

Transaction Report:

Dear Dr Coronel,

Re: JP-TR-2024-286978 "Cellular and Molecular Cross-talk in Atrial Fibrillation: The Role of Non-cardiomyocytes in Creating an Arrhythmogenic Substrate" by Zhenyu Dong, Ruben Coronel, and Joris R. de Groot

Thank you for submitting your manuscript to The Journal of Physiology. It has been assessed by a Reviewing Editor and by 2 expert referees and we are pleased to tell you that it is potentially acceptable for publication following satisfactory major revision.

Please address all the points raised and incorporate all requested revisions or explain in your Response to Referees why a change has not been made. We hope you will find the comments helpful and that you will be able to return your revised manuscript within 2 months. If your article is for a Special Issue, please note that we require your revised version within 2 months (rather than 9 months) in order to keep the Special issue on track. If you require longer than this, please contact journal staff: jp@physoc.org. Please note that this letter does not constitute a guarantee for acceptance of your revised manuscript.

ABSTRACT FIGURES: Authors are expected to use The Journal's premium BioRender account to create/redraw their Abstract Figures. Information on how to access this account is here:

<https://physoc.onlinelibrary.wiley.com/journal/14697793/biorender-access>.

REVISION CHECKLIST:

IMPORTANT POINTS TO NOTE WHEN REVISING YOUR MANUSCRIPT:

LANGUAGE EDITING AND SUPPORT FOR PUBLICATION: If you would like help with English language editing, or other article preparation support, Wiley Editing Services offers expert help, including English Language Editing, as well as translation, manuscript formatting, and figure formatting at www.wileyauthors.com/eoo/preparation. You can also find resources for Preparing Your Article for general guidance about writing and preparing your manuscript at www.wileyauthors.com/eoo/prepresources.

We look forward to receiving your revised submission.
If you have any queries, please reply to this email and we will be pleased to advise.

Yours sincerely,

Bjorn Knollmann
Senior Editor
The Journal of Physiology

EDITOR COMMENTS

Your paper has been reviewed by two experts in the field, who both felt it is an informative and comprehensive manuscript focused on an important and timely topic, which will make it a valuable addition to the literature. While they agree it is an excellent paper, they both had suggestions that would improve its overall quality, as well as a few elements that may deserve further attention. Please revise the manuscript accordingly, including a point-by-point response to the reviewers' suggestions.

Senior Editor:

I concur with the reviewing editor.

REFEREE COMMENTS

Referee #1:

This is an informative review on a timely topic. I have a few suggestions which might help to further improve the context of this manuscript.

Major comments

- Cell diversity in the cardiovascular system is complex including additional aspects not addressed here. Please refer to a recent review article (PMID: 36744541)
- Endothelial-mesenchymal transition a mechanism of profibrotic remodeling needs mentioning (e.g. PMID: 35775488)
- Adipocytes in AF: a recent review on the role of PAT in AF development should be cited (PMID: 35689487).
- Coagulation system in AF: the putative pleiotropic antiarrhythmic actions were summarized elsewhere (PMID: 31054953). Conversely, there is evidence that thrombin could act specifically on PAR type-4 receptors to activate inflammatory signaling, particularly in cardiac fibroblasts (PMID: 31912235). Thus, both the putative pro- and antiarrhythmic actions of OACs should be discussed.
- Inflammation in AF: Please stress to the readers that inflammatory signaling via the NLRP3 inflammasome system in atrial cardiomyocytes could promote AF in patients (PMID: 32762493) and that key AF-promoting comorbidities engage the same system to promote AF (obesity: PMID: 33523143; chronic kidney disease: PMID: 37581942; gut microbiota: PMID: 34550344).
- IL Family: IL-1beta causes proarrhythmic SR Ca²⁺ release events in atrial cardiomyocytes which might cause cellular triggered activity (PMID: 32762493). IL-6 trans-signaling causes AF (PMID: 37633428), with selective IL-6 inhibition preventing AF development. These studies should be discussed.
- Page 17: add thrombin actions via PAR4 in heart, potentially causing remodeling (PMID: 31912235)

Minor comments

Page 18: Yao et al., 2018b should be Yao et al., 2018

Page 29: McCaulley et al. 2024 is mentioned twice. Please fix.

Referee #2:

"Cellular and Molecular Cross-talk in Atrial Fibrillation: The Role of Non-cardiomyocytes in Creating an Arrhythmogenic Substrate" by Dong, Coronel and de Groot is a comprehensive review investigating the current understanding of how non-cardiomyocytes within the heart modulate cardiac electrophysiology, destabilizing the electrical landscape of the atria and promoting the pathophysiological changes with underlying atrial fibrillation (AF). The authors investigate the interactions of the myocardium with fibroblasts/myofibroblasts, adipose tissue, immune cells and elements in the coagulation cascade. These non-muscle cells and elements are associated with inflammation, fibrosis, unhealthy secretosomes and modulation of cardiomyocyte resting membrane and action potential characteristics which predispose the atria to fibrillation.

The review is well researched and includes excellent definitions of cross-talk, expanding the common views of chemical and gap junction-mediated to include other forms of direct cell coupling and anatomical cross-talk. This platform allows for a richer understanding of the important interactions between muscle and other cells in the heart, in health and in illness.

The focus on direct and indirect cross-talk which is primarily chemical while placing this element in the context of the highly complex 'eco-system' within the myocardium is an effective way to present this important mechanistic understanding.

Impact on the area of research

The review is a valuable addition to the literature on the pathophysiological mechanisms underpinning AF, without trying to tackle the very difficult task of understanding the progression from paroxysmal to persistent AF (though AF progression is mentioned). Addition of the influence of coagulation, a key issue in an acute coronary syndrome, is novel and an important dimension to introduce to the discussion. The latest work is summarised, allowing the reader to appreciate the current understanding of the many interactions.

Insight into physiological mechanisms in this field

The authors provide a thorough investigation into direct and indirect chemical cross-talk which contributes to the pathogenesis of AF in the fibrotic, inflamed myocardium. The presentation of the many elements is solid but seems to lack integration - in this reviewer's opinion, rethinking (and rewriting) the information so that the presentation is thematic more than categorical would be an improvement. However, it is clear that to do so would be challenging and while this reviewer offers this as a major suggestion, it may prove to be too difficult due to the nature of the many categories of interactions and lists of effects. A critical reader will take the categorical organisation of the information and extract thematic ideas from it.

Validity of the conclusions

The conclusions are appropriate and based on the information provided. The inclusion of the need for novel therapeutic strategies, which might be developed to target the cross-talk, is appropriate and provides the reader with a sense of where future research may translate into improved patient management. The final summary (lines 495 - 500) focuses on epicardial adipose tissue (EAT) and obesity, where AF is a common comorbidity. However, this ignores the sections on (myo)fibroblasts and fibrosis, as well as immune cells and the coagulation cascade. This reviewer recommends expanding this section to include inflammation in obesity and also mention coronary artery disease and coagulation. These elements would more completely summarise the topics in the review, and would highlight the most novel addition: the influence of coagulation on myocardial electrophysiology.

Major Points:

1. The categorical organisation of the review makes thematic interpretation more difficult. If the review were re-written to consider themes such as autocrine/paracrine hormonal cross-talk with inclusive descriptions of myocytes, (myo)fibroblasts, adipocytes, white-blood cells and agents released during blood clotting, the key mechanistic effects might be more apparent. The categorical organisation according to cell type is also effective but does not allow the full integration across multiple cell types to be as easily understood

2. More descriptive legends are used to describe tables 1 and 2 would be advantageous for readers who do not read the accompanying text. Also suggest that these tables be rotated to landscape orientation so that they are larger and easier to read. It would be clearer in Table 2 to provide references as "↑ (1)" rather than "↑ 1". Also, please re-order so that reference 24 is 11, as would be correct for ascending numerical according to appearance. In the figures, the term "(myo)fibroblast" may be more accurate and informative than "fibroblast". Are references needed in the figures? Figure 2 would be improved by removing "presynaptic" and "postsynaptic" cell (autocrine panel). Furthermore, for the endocrine and paracrine panels, having the green cell look less like an axon terminus would be helpful to prevent confusion with neuroendocrine cross-talk, which was only mentioned in the review. Figures such as figure 3 have very small font and may be better rotated to landscape view to allow a larger presentation

3. In describing the electrophysiology of cardiac fibroblasts, some of the works from the Giles lab (and if appropriate, co-authored publications by Giles, Smith and Kohl) could be included to provide a more complete picture. For example, in contrast to Rook et al. 1992, Kamkin et al. 1999 and Bentley et al. 2014, Chilton et al. 2005 (doi:10.1152/ajpheart.01220.2004) recorded the inwardly rectifying potassium channel Kir2.1 currents in (myo)fibroblasts and resting potentials closer to the equilibrium potential for potassium. It seems worth including this information as well as evidence that other fibroblasts are more depolarised at rest, to provide a more complete picture of possible resting membrane potentials in (myo)fibroblasts and the effect that electrotonic coupling of myocytes to these cells may have on myocyte resting membrane potential, as per MacCannel et al. 2007

4. The text includes numerous grammatical and format errors. For example, on line 41, "approximately >5.6 million" is not grammatically correct. The correct wording would be "approximately 5.6 million" or ">5.6 million". Similarly, line 284 "These mice also exhibit longer APD, with APD75 and APD90 (14% and 28%, respectively)" should be more clearly written; e.g., "Cardiomyocytes from these mice also exhibited longer action potential durations (APD), with APD75 and APD90 increased by 14 and 28% respectively". Multiple places lack a space between a word and opening bracket, as with line 346 "... containing protein 3(NLRP3)" rather than "... containing protein 3 (NLRP3)" and line 351 ("macrophages to the atria(Dobrey et al. 2013)" rather than "macrophages to the atria (Dobrey et al. 2013)". There are numerous typos, including (I assume) in the list of authors (line 3, "Ruben Coronel R" - is "R" meant to be an abbreviation for Dr Coronel's middle name?), misspelling on line 479 ("complex" not "complex"); lack of capital letters (e.g., line 138, Small Mothers Against Decapentaplegic (SMAD) and the many proper nouns in the Funding section on lines 540 to 545), incorrect predictive text (e.g., line 275-276 "which and increased AF risk" rather than "with an increased AF risk", presumably; line 540 "...advancement -itea4" (what is "itea4?)" and line 542 "...0118 embrace." (is "embrace" correct?)), incorrect spacing (e.g., line 161: 2.1. 2.4 rather than 2.1.2.4), missing full stops (lines 207, 219 and 254), extra comma (line 258) or words (lines 332, 425, 435) and incomplete sentences (e.g., line 120)

5. Both in-text and the list of references require checking. References are not always provided in the text (e.g., lines 86-87, 176-178, 183, 224, 368, 371, 459-461, 485-488, and for the unreferenced information in Table 1). References in text have errors (lines 103-104, de Baker and De Baker - please be consistent if this format is not correct; line 160 "Ngu et al. 2014" in text but Ngu is the solo author in the list of references, line 864; lines 186 -187, "Hatem & Sanders" rather than "Hatem and Sanders") as does the list of references (lines 585 - 587 "Ausma et al. 2001" does not appear to be cited in the text; lines 592 - 598, "de Baker et al. 1993" is repeated; line 709 the reference Hall et al. 2021 needs to be included as it is cited in the text in Table 1 but missing from the list of references; line 712 - 714 "He et al. 2011" was not cited in the text; lines 826 - 831 "McCauley et al. 2024a and 2024b are the same paper, referred to only as "...2024" in the text; line 866 - the citation Oikonomou and Antonaidis 2019 needs to be added as it was cited in text on line 171; line 888 the citation Rushd et al. 2023 needs to be added as it was cited in Table 2; line 972 "Yao C et al. (2018)" is referred to as "2018b" in the text on line 347

Minor Points:

1. Please check abbreviations - write out at first use, use consistently there-after and consider including a list of abbreviations

2. Please ensure that the studies described in the review are in the past tense

3. Please use the term "cardiomyocyte" consistently

END OF COMMENTS

Response to the reviewing editor and Senior Editor

Editor comments:

Your paper has been reviewed by two experts in the field, who both felt it is an informative and comprehensive manuscript focused on an important and timely topic, which will make it a valuable addition to the literature. While they agree it is an excellent paper, they both had suggestions that would improve its overall quality, as well as a few elements that may deserve further attention. Please revise the manuscript accordingly, including a point-by-point response to the reviewers' suggestions.

Senior Editor:

I concur with the reviewing editor.

We appreciate the editors' positive assessment of our paper. We have carefully addressed all reviewers' comments and uploaded two versions of the revised manuscript. Below, we provide a point-by-point response to each comment, with references to the corresponding page and line numbers in the clean version. In the tracked-changes version, the corrections made in response to Reviewer 1 are highlighted in **green**, while those for Reviewer 2 are in **yellow**. We also made additional changes to the main text for clarification. These changes are highlighted in **blue** in the tracked-changes version.

Response to Referee #1

This is an informative review on a timely topic. I have a few suggestions which might help to further improve the context of this manuscript.

Thank you for your positive report. We have addressed your comments point by point below, referencing the relevant page and line numbers in the manuscript. Additionally, we have highlighted all changes in **green** in the tracked-changes version.

However, due to constraints (word count, number of references), we were unable to cover all aspects of crosstalk in detail. We have concentrated our considerations on those alluded to in Figure 2 (red arrows) and that directly interfere with cardiomyocyte function. Thus, while we briefly mention some of the intriguing points you raised, we regret that we cannot discuss them more thoroughly. In the introduction, we clarify the focus of the review.

Major Comments

1. Comment: Cell diversity in the cardiovascular system is complex, including additional aspects not addressed here. Please refer to a recent review article (PMID: 36744541).

Response:

Thank you for your suggestion. We have supplemented our discussion to address these additional aspects, citing the recommended review article (Grandi et al., 2023). This addition can be found on page 3, lines 54-57, as follows:

“Other non-cardiomyocytes affecting the structural and functional properties of the heart that may lead to AF include (myo)fibroblasts, adipocytes, inflammatory cells, vascular smooth muscle cells, endothelial cells, and neurons (Grandi et al., 2023).”

2. Comment: Endothelial-mesenchymal transition as a mechanism of profibrotic remodeling needs mentioning (PMID: 35775488).

Response: We agree with the reviewer and have now included endothelial-mesenchymal transition as a key mechanism of profibrotic remodeling, citing Saljic et al., 2022. This addition has been made on page 3, lines 63-65 as follows:

“Endothelial-mesenchymal transition has been identified as a mechanism underlying profibrotic remodeling in AF (Saljic et al., 2022; van den Berg et al., 2021).”

3. Comment: Adipocytes in AF: a recent review on the role of PAT in AF development should be cited (PMID: 35689487).

Response: The role of pericardial adipose tissue (PAT) in AF development has been included, with citation to the recommended review (Gawałko et al., 2023), together with a

more recent paper on the subject (Pyman et al., 2024). This has been added on page 7, lines 163-165:

“Adipocytes in the heart are primarily located within the epicardial adipose tissue (EAT), which is the visceral adipose tissue between the myocardium and the epicardium (Antonopoulos & Antoniadis, 2017; Gawalko et al., 2023; Pyman et al., 2024).”

4. Comment: Coagulation system in AF: the putative pleiotropic antiarrhythmic actions were summarised elsewhere (PMID: 31054953). Conversely, there is evidence that thrombin could act specifically on PAR type-4 receptors to activate inflammatory signaling, particularly in cardiac fibroblasts (PMID: 31912235). Thus, both the putative pro- and antiarrhythmic actions of OACs should be discussed.

Response:

We have incorporated references to (Fender et al., 2020) and (Fender et al., 2019), which can be found on page 9 (lines 222–223) and page 15 (lines 434–435).

“However, this perspective has recently been challenged (Brambatti et al., 2014; de Jong et al., 2017; Fender et al., 2019).”

“Additionally, PAR4 promoted inflammatory responses through the NLRP3 inflammasome (Fender et al., 2020) (e in Figure 2).”

5. Comment: Inflammation in AF: Please stress to the readers that inflammatory signaling via the NLRP3 inflammasome system in atrial cardiomyocytes could promote AF in patients (PMID: 32762493). Also, key AF-promoting comorbidities engage the same system to promote AF (obesity: PMID: 33523143; chronic kidney disease: PMID: 37581942; gut microbiota: PMID: 34550344).

Response:

We have now emphasized the role of the NLRP3 inflammasome system in atrial cardiomyocytes and its connection to AF-promoting comorbidities (obesity, CKD, and gut microbiota). The relevant references (Heijman et al., 2020; Scott et al., 2021; Song et al., 2023; Gawalko et al., 2022) have been added to the discussion on page 12, lines 331-332, 335 – 338:

“Increased NLRP3 activity in atrial cardiomyocytes was linked with atrial fibrosis and higher AF inducibility in animal models (Yao et al., 2018; Heijman et al., 2020).”

“Additionally, several AF-promoting comorbidities that were demonstrated to be closely associated with AF progression also contributed to AF occurrence through NLRP3 inflammasome activation, including obesity (Scott et al., 2021), chronic kidney disease (Song et al., 2023), and gut microbiota (Gawalko et al., 2022).”

6. Comment: IL Family: IL-1beta causes proarrhythmic SR Ca²⁺ release events in atrial cardiomyocytes, which might cause cellular triggered activity (PMID: 32762493). IL-6 trans-signaling causes AF (PMID: 37633428), with selective IL-6 inhibition preventing AF development. These studies should be discussed.

Response: We have mentioned the roles of IL-1beta and IL-6's in AF development, highlighting their contributions to proarrhythmic signalling and the potential for therapeutic targeting, citing Heijman et al., 2020; Li et al., 2023. This addition has been included on page 11 (lines 290–291) and page 11 (lines 296–298):

“IL-1 β also increased spontaneous diastolic sarcoplasmic reticulum Ca²⁺ release in cardiomyocytes and decreased the expression of Ca²⁺-handling proteins (Monnerat et al., 2016; Szekely & Arbel, 2018; Heijman et al., 2020).”

“Elevated IL-6 signalling levels in patients were positively associated with AF, and selective blockade of IL-6 signalling reduced AF inducibility (Li et al., 2023).”

7. Comment: Page 17: Add thrombin actions via PAR4 in the heart, potentially causing remodeling (PMID: 31912235).

Response: We have added the discussion on thrombin's actions via PAR4 in the heart, particularly focusing on its potential role in remodelling (Fender et al., 2020). This is addressed on page 9, lines 247-248:

“Research by Fender et al. suggested that thrombin promoted cardiac remodelling through Protease-Activated Receptor 4 (PAR4) (Fender et al., 2020).”

Minor Comments

1. Comment: Page 18: Yao et al., 2018b should be Yao et al., 2018.

Response: This has been corrected to 'Yao et al., 2018' as suggested.

2. Comment: Page 29: McCaulley et al. 2024 is mentioned twice. Please fix.

Response: We have corrected this duplicate citation.

Response to Referee #2

"Cellular and Molecular Cross-talk in Atrial Fibrillation: The Role of Non-cardiomyocytes in Creating an Arrhythmogenic Substrate" by Dong, Coronel and de Groot is a comprehensive review investigating the current understanding of how non-cardiomyocytes within the heart modulate cardiac electrophysiology, destabilizing the electrical landscape of the atria and promoting the pathophysiological changes with underlying atrial fibrillation (AF). The authors investigate the interactions of the myocardium with fibroblasts/myofibroblasts, adipose tissue, immune cells and elements in the coagulation cascade. These non-muscle cells and elements are associated with inflammation, fibrosis, unhealthy secretosomes and modulation of cardiomyocyte resting membrane and action potential characteristics which predispose the atria to fibrillation.

The review is well researched and includes excellent definitions of cross-talk, expanding the common views of chemical and gap junction-mediated to include other forms of direct cell coupling and anatomical cross-talk. This platform allows for a richer understanding of the important interactions between muscle and other cells in the heart, in health and in illness.

The focus on direct and indirect cross-talk which is primarily chemical while placing this element in the context of the highly complex 'eco-system' within the myocardium is an effective way to present this important mechanistic understanding.

Impact on the area of research

The review is a valuable addition to the literature on the pathophysiological mechanisms underpinning AF, without trying to tackle the very difficult task of understanding the progression from paroxysmal to persistent AF (though AF progression is mentioned). Addition of the influence of coagulation, a key issue in an acute coronary syndrome, is novel and an important dimension to introduce to the discussion. The latest work is summarised, allowing the reader to appreciate the current understanding of the many interactions.

Insight into physiological mechanisms in this field

The authors provide a thorough investigation into direct and indirect chemical cross-talk which contributes to the pathogenesis of AF in the fibrotic, inflamed myocardium. The presentation of the many elements is solid but seems to lack integration - in this reviewer's opinion, rethinking (and rewriting) the information so that the presentation is thematic more than categorical would be an improvement. However, it is clear that to do so would be challenging and while this reviewer offers this as a major suggestion, it may prove to be too difficult due to the nature of the many categories of interactions and lists of effects. A critical reader will take the categorical organisation of the information and extract thematic ideas from it.

Validity of the conclusions

The conclusions are appropriate and based on the information provided. The inclusion of the need for novel therapeutic strategies, which might be developed to target the cross-talk, is appropriate and provides the reader with a sense of where future research may translate into improved patient management. The final summary (lines 495 - 500) focuses on epicardial adipose tissue (EAT) and obesity, where AF is a common comorbidity. However, this ignores the sections on (myo)fibroblasts and fibrosis, as well as immune cells and the coagulation cascade. This reviewer recommends expanding this section to include inflammation in obesity and also mention coronary artery disease and coagulation. These elements would more completely summarise the topics in the review, and would highlight the most novel addition: the influence of coagulation on myocardial electrophysiology.

Response:

Thank you for your detailed comments and the positive report. We have addressed your comments point by point below, referencing the relevant page and line numbers in the manuscript. Additionally, we have highlighted all changes in **yellow** in the tracked-changes version. We also made additional changes to the main text for clarification. These changes are highlighted in **blue** in the tracked-changes version.

Due to constraints (word count, number of references), we were unable to cover all aspects of cross-talk in detail. We have concentrated our considerations on those alluded to in Figure 2 (red arrows) and that directly interfere with cardiomyocyte function. Thus, while we briefly mention some of the intriguing points you raised, we regret that we cannot discuss them more thoroughly. In the introduction, we clarify the focus of the review.

We agree that our conclusion was previously oriented toward adiposity, and we have now included references to multiple other non-cardiomyocyte cell types. The revised sentences at page 16, lines 458–461 are as follows:

“Our review underscores how a complex network of cellular cross-talk involving inflammation, fibrosis, EAT, and the coagulation cascade collectively contributes to an arrhythmogenic substrate for AF.”

We have also mentioned coronary artery disease and coagulation on page 8 (lines 211–212) and page 15 (lines 432–434), as follows:

“The coagulation system plays a critical role in cardiovascular diseases, including coronary heart disease and arrhythmias (Coronel et al., 1997; Watson et al., 2009).”

“Through this mechanism, EAT promoted the formation and stabilisation of atherosclerotic plaques (Kohler & Grant, 2000).”

Major Points:

1. Comment

The categorical organisation of the review makes thematic interpretation more difficult. If the review were re-written to consider themes such as autocrine/paracrine hormonal cross-talk with inclusive descriptions of myocytes, (myo)fibroblasts, adipocytes, white-blood cells and agents released during blood clotting, the key mechanistic effects might be more apparent. The categorical organisation according to cell type is also effective but does not allow the full integration across multiple cell types to be as easily understood."

Response:

We appreciate the reviewer's comment on the apparent lack of integration and the suggestion to reorganize the review to yield a more thematic presentation. This is indeed a challenge, and the reviewer rightfully acknowledges that categorical organization according to cell type is also effective. Because we focus more on paracrine mechanisms, the effects of secreted signalling molecules are difficult to categorize. We fear that changing the organization in this sense would lead to another sense of disorganization. We, therefore, addressed this point by organizing the information in the table into a more thematic order (direct and indirect cross-talk). We hope that this will yield a categorical as well as a thematic approach and will help the reader to switch between the two perspectives. Furthermore, we have chosen to restrict ourselves to the immediate interactions with the cardiomyocytes (see Figure 2, red arrows) and feel that the thematic approach is more suited to address this 'limitation'. The restriction is now made more explicit in the introduction.

2. Comment:

More descriptive legends are used to describe tables 1 and 2 would be advantageous for readers who do not read the accompanying text.

Response:

We have expanded the legends for Tables 1 and 2 to include more descriptive details. These additions can be found on page 17, lines 466-468 as follows:

"Table 1 Types and examples of cellular and molecular cross-talk in atrial fibrillation"

"Table 2 Effects of representative non-cardiomyocytes on structural and electrophysiological changes in cardiomyocytes during atrial fibrillation"

Also suggest that these tables be rotated to landscape orientation so that they are larger and easier to read.

Both tables have been rotated to a landscape orientation for improved readability.

It would be clearer in Table 2 to provide references as '↑ (1)' rather than '↑ 1'.

References in Table 2 have been updated to the format "↑ (1)".

Conduction velocity	APD	ERP	RMP
↓ (18)	↓ (20)		
	↓ (21)		↑ (26)
	↑ (22)		↑ (22)
↓ (19)	↑ (12)		

Also, please re-order so that reference 24 is 11, as would be correct for ascending numerical according to appearance.

We have re-ordered the table numbers to ensure the correct ascending order according to appearance.

In the figures, the term '(myo)fibroblast' may be more accurate and informative than 'fibroblast'.

We have replaced the term "fibroblast" with "(myo)fibroblast" in the figures for greater accuracy.

Are references needed in the figures?

Regarding figure references, we have added them at page 17, lines 474–478, as follows:

“Figure 1 Different modes of cell cross-talk include indirect methods, such as autocrine, endocrine, and paracrine signalling, and direct methods, including nanotubes, mechanical interactions, and gap junctions. The right panel depicts the interactions between signalling molecules, using Transforming Growth Factor-beta (TGF-β), Angiotensin II (Ang II), and Connective Tissue Growth Factor (CTGF) as examples (Tsai et al., 2011; Chen et al., 2000).”

Figure 2 would be improved by removing 'presynaptic' and 'postsynaptic' cell (autocrine panel).

In Figure 2, we have removed the "presynaptic" and "postsynaptic" labels in the autocrine panel to avoid confusion.

Furthermore, for the endocrine and paracrine panels, having the green cell look less like an axon terminus would be helpful to prevent confusion with neuroendocrine cross-talk, which was only mentioned in the review.

The green cell in the endocrine and paracrine panels has been redesigned to prevent it from being misinterpreted as an axon terminus. Please refer to the updated figure above.

Figures such as figure 3 have very small font and may be better rotated to landscape view to allow a larger presentation.

Figure 3 has been revised into an abstract figure to achieve better presentation(bigger). To ensure clarity, we removed the smaller fonts and focused solely on conveying the most essential information

3. Comment:

In describing the electrophysiology of cardiac fibroblasts, some of the works from the Giles lab (and if appropriate, co-authored publications by Giles, Smith and Kohl) could be included to provide a more complete picture. For example, in contrast to Rook et al. 1992, Kamkin et al. 1999 and Bentley et al. 2014, Chilton et al. 2005 (doi:10.1152/ajpheart.01220.2004) recorded the inwardly rectifying potassium channel Kir2.1 currents in (myo)fibroblasts and resting potentials closer to the equilibrium potential for potassium. It seems worth including this information as well as evidence that other fibroblasts are more depolarised at rest, to provide a more complete picture of possible resting membrane potentials in (myo)fibroblasts and the effect that electrotonic coupling of

myocytes to these cells may have on myocyte resting membrane potential, as per MacCannel et al. 2007.

Response:

We appreciate this suggestion and have incorporated the works from Giles et al. We have also cited the study by Chilton et al. (2005) to provide a more complete picture of the electrophysiology of cardiac fibroblasts, and we noted that there is indeed considerable heterogeneity in the research, as highlighted by Simon-Chica et al. (2023) (page 13, line 345-349).

Changes made as follows:

“Chilton et al. demonstrated that Kir2.1 in fibroblasts directly affects the resting membrane potential, bringing it closer to the potassium equilibrium potential (Chilton et al., 2005). However, the resting membrane potential in fibroblasts exhibits notable heterogeneity (Simon-Chica et al., 2023).”

4. Comments:

The text includes numerous grammatical and format errors. For example, on line 41, 'approximately >5.6 million' is not grammatically correct. The correct wording would be 'approximately 5.6 million' or '>5.6 million'.

Thank you. Page 3, lines 39-41: 'Approximately >5.6 million' has been replaced with a more accurate figure. We selected the updated reference Noubiap et al., 2024, so the current number is now over 10.55 million:

"Atrial fibrillation (AF) is the most common arrhythmia and is currently estimated to affect over 10.55 million patients in the United States and 11.7 million in the European Union (Noubiap et al., 2024; Krijthe et al., 2013).”

Similarly, line 284 'These mice also exhibit longer APD, with APD75 and APD90 (14% and 28%, respectively)' should be more clearly written; e.g., 'Cardiomyocytes from these mice also exhibited longer action potential durations (APD), with APD75 and APD90 increased by 14 and 28% respectively'.

Thank you. We have revised the sentence on page 10, lines 277–278 as follows:

"Cardiomyocytes from these mice also exhibited a prolonged APD, with APD75 and APD90 increased by 14% and 28%, respectively (London et al., 2003).”

Multiple places lack a space between a word and opening bracket, as with line 346 '... containing protein 3(NLRP3)' rather than '... containing protein 3 (NLRP3)' and line 351 'macrophages to the atria(Dobrey et al. 2013)' rather than 'macrophages to the atria (Dobrey et al. 2013)'.

The text has been corrected as follows:

On page 12, line 329:

“...containing protein 3 (*NLRP3*)...”

On page 12, line 334:

“...to the atria (*Dobrev et al., 2023*)”

We have replaced similar occurrences throughout the text.

There are numerous typos, including (I assume) in the list of authors (line 3, 'Ruben Coronel R' - is 'R' meant to be an abbreviation for Dr Coronel's middle name?), misspelling on line 479 ('complex' not 'complex');

Line 3 (author list): The "R" in "Ruben Coronel R" is not his middle name and has been removed.

Page 16, Line 444: Corrected the typo from "compex" to "complex."

lack of capital letters (e.g., line 138, Small Mothers Against Decapentaplegic (SMAD) and the many proper nouns in the Funding section on lines 540 to 545), incorrect predictive text (e.g., line 275-276 'which and increased AF risk' rather than 'with an increased AF risk', presumably;

Proper capitalization has been applied to all specialized terms. However, for better readability, the rarely used ‘Small Mothers Against Decapentaplegic’ has been removed.

Page 10, Lines 268-270:

The sentence has been updated to: “*This results in loss of cardiomyocyte function and decreased cardiac contractility, which increases AF risk (Dobrev et al., 2023).*”

line 540 '...advancement -itea4' (what is 'itea4?) and line 542 '...0118 embrace.' (is 'embrace' correct?),

Corrected the phrase "advancement -itea4" and clarified the intended meaning.

Page 18, Line 517: We changed ‘embrace’ to uppercase and provided the full name: *Information Technology for European Advancement (ITEA4)*

Page 18, Line 519: We changed ‘embrace’ to uppercase and provided the full name: *Electrophysiological and Molecular Biomarkers in Atrial Fibrillation (EmBRACE)*

incorrect spacing (e.g., line 161: 2.1. 2.4 rather than 2.1.2.4),

Page 6, Line 156: Corrected spacing to "2.1.2.4" from "2.1. 2.4."

missing full stops (lines 207, 219 and 254),

We have added the missing full stops and thoroughly revised the punctuation throughout this manuscript.

extra comma (line 258) or words (lines 332, 425, 435) and incomplete sentences (e.g., line 120)."

Corrected commas and removed extra words to improve clarity.

The sentence in Line 120 has been removed to improve logical coherence.

5. Comment:

"Both in-text and the list of references require checking. References are not always provided in the text (e.g., lines 86-87, 176-178, 183, 224, 368, 371, 459-461, 485-488, and for the unreferenced information in Table 1).

We apologize for the confusion in the reference list. We have thoroughly revised and checked the references. References that were missing from the text have been added, including, but not limited to, those suggested by the reviewers.

References in text have errors (lines 103-104, de Baker and De Baker - please be consistent if this format is not correct;

Line 100-101: Inconsistent reference formatting (e.g., "de Baker and De Baker") has been corrected to "de Bakker et al." throughout.

line 160 'Ngu et al. 2014' in text but Ngu is the solo author in the list of references, line 864;

Thank you. The incorrect citation of "Ngu et al. 2014" has been corrected, and Ngu has been properly referenced in the text and list.

Page 6, Line 154-155: *TIMP-2 became inhibitory to this transformation process (Ngu et al., 2014).*

Page 30, Line 829-832: *Ngu JMC, Teng G, Meijndert HC, Mewhort HE, Turnbull JD, Stetler-Stevenson WG & Fedak PWM (2014). Human cardiac fibroblast extracellular matrix remodeling: dual effects of tissue inhibitor of metalloproteinase-2. Cardiovascular Pathology, 23(6), 335–343.*

lines 186 -187, 'Hattem & Sanders' rather than 'Hattem and Sanders') as does the list of references

Thank you for the feedback. We have updated page 7, lines 171, to "Hattem & Sanders, 2014."

Lines 180-181 have been revised as follows:

"This is consistent with findings in mouse models (Nalliah et al., 2020; Hattem & Sanders, 2014)."

lines 585 - 587 'Ausma et al. 2001' does not appear to be cited in the text;

Thank you for pointing this out. We have removed “Ausma et al. 2001” as it was incorrectly included.

lines 592 - 598, 'de Baker et al. 1993' is repeated;

The repeated citation of "de Baker et al. 1993" has been removed, and the correct version remains. See above.

line 709 the reference Hall et al. 2021 needs to be included as it is cited in the text in Table 1 but missing from the list of references;

Line 709: "Hall et al. 2021" has been replaced with other references in Table 1 to provide more appropriate examples.

Example
Intercellular coupling between myocytes and myofibroblasts influence the electrical activity of the myocyte. (Chilton et al., 2007)
Myofibroblast tension on cardiomyocytes may impair conduction via mechanosensitive channel activation (Thompson et al., 2011)
Fibroblasts can synchronize electrical activity across tissue via electrotonic interactions and membrane nanotubes. (Gaudesius et al., 2003)
Adipocytes secreting MicroRNA that impact cardiomyocytes(Ernault et al., 2023)
Fibroblasts producing TGF- β to enhance fibrosis (Ramos-Mondragón et al., 2008)
Myeloperoxidase released by immune cells enter the bloodstream and subsequently affect cardiomyocytes. (Rudolph et al., 2010)

line 712 - 714 'He et al. 2011' was not cited in the text;

Thank you for the feedback. We have now included “He et al., 2011” at page 14, line 402-403. The revised text is as follows:

“In vitro and in vivo (Gaudesius et al., 2003; Gerdes & Carvalho, 2008; He et al., 2011) (Figure 1).”

lines 826 - 831 'McCauley et al. 2024a and 2024b are the same paper, referred to only as '...2024' in the text;

The duplicate entries for "McCauley et al. 2024a and 2024b" have been combined, and the correct reference is now consistently referred to as "*McCauley et al. 2024*".

line 866 - the citation Oikonomou and Antonaides 2019 needs to be added as it was cited in text on line 171;

"Oikonomou and Antonaides 2019" has been removed.

line 888 the citation Rushd et al. 2023 needs to be added as it was cited in Table 2;

"Rushd et al. 2023" has been changed to "Al-Shama et al., 2023". Page 20, Line 533-536: *"Al-Shama RFM, Ernault AC, Meulendijks ER, Fabrizi B, van Amersfoort SCM, Boender AR, Coronel R, Boukens BJD & de Groot JR (2023). Myeloperoxidase causes both arrhythmogenic structural and electrical remodelling in neonatal rat ventricular myocyte monolayers. Europace, 25(Supplement_1), eoad122.016."*

line 972 'Yao C et al. (2018)' is referred to as '2018b' in the text on line 347)."

The erroneous citation of "Yao C et al. (2018b)" has been corrected to "Yao et al. (2018)" throughout.

Minor Points:

1. Comment:

Please check abbreviations - write out at first use, use consistently there-after and consider including a list of abbreviations.

Response:

All abbreviations have been reviewed to ensure that they are spelled out upon first mention and used consistently thereafter. The list of abbreviations is provided in Lines 483–507.

2. Comment:

Please ensure that the studies described in the review are in the past tense.

Response:

We have revised the manuscript to ensure that all studies described are consistently written in the past tense.

3. Comment:

Please use the term 'cardiomyocyte' consistently.

Response:

The term "cardiomyocyte" has been reviewed and is now used consistently throughout the manuscript to avoid any confusion,

Dear Dr Coronel,

Re: JP-TR-2025-286978R1 "**Cellular and Molecular Cross-talk in Atrial Fibrillation: The Role of Non-cardiomyocytes in Creating an Arrhythmogenic Substrate**" by Zhenyu Dong, Ruben Coronel, and Joris R. de Groot

We are pleased to tell you that your paper has been accepted for publication in The Journal of Physiology.

Authors should note that it is too late at this point to offer corrections prior to proofing. Major corrections at proof stage, such as changes to figures, will be referred to the Editors for approval before they can be incorporated. Only minor changes, such as to style and consistency, should be made at proof stage. Changes that need to be made after proof stage will usually require a formal correction notice.

Yours sincerely,

Bjorn Knollmann
Senior Editor
The Journal of Physiology

P.S. - You can help your research get the attention it deserves! Check out Wiley's free Promotion Guide for best-practice recommendations for promoting your work at www.wileyauthors.com/eoo/guide. You can learn more about Wiley Editing Services which offers professional video, design, and writing services to create shareable video abstracts, infographics, conference posters, lay summaries, and research news stories for your research at www.wileyauthors.com/eoo/promotion.

IMPORTANT NOTICE ABOUT OPEN ACCESS: To assist authors whose funding agencies mandate public access to published research findings sooner than 12 months after publication, The Journal of Physiology allows authors to pay an Open Access (OA) fee to have their papers made freely available immediately on publication.

You can check if your funder or institution has a Wiley Open Access Account here: <https://authorservices.wiley.com/author-resources/Journal-Authors/licensing-and-open-access/open-access/author-compliance-tool.html>.

EDITOR COMMENTS

Reviewing Editor:

The authors have satisfied all recommendations of the authors.

Senior Editor:

The article is now acceptable for publication. I thank the authors for contributing this thought-provoking and thoroughly researched piece.

REFeree COMMENTS

Referee #1:

Thank you for the responsive revision. No further comments.